# A coopetition-driven strategy of parallel/ perpendicular aromatic stacking enabling metastable supramolecular polymerization

Zhao Gao [1,3], Xuxu Xie[1,3], Juan Zhang[1], Wei Yuan[2], Hongxia Yan[1] & Wei Tian [1] ✉

Metastable supramolecular polymerization under kinetic control has recently been recognized as a closer way to biosystem than thermodynamic process. While impressive works on metastable supramolecular systems have been reported, the library of available non-covalent driving modes is still small and a simple yet versatile solution is highly desirable to design for easily regulating the energy landscapes of metastable aggregation. Herein, we propose a coopetition-driven metastability strategy for parallel/perpendicular aromatic stacking to construct metastable supramolecular polymers derived from a class of simple monomers consisting of lateral indoles and aromatic core. By subtly increasing the stacking strength of aromatic cores from phenyl to anthryl, the parallel face-to-face stacked aggregates are competitively formed as metastable products, which spontaneously transform into thermo-dynamically favorable species through the cooperativity of perpendicular edge-to-face stacking and parallel offset stacking. The slow kinetic-to-thermodynamic transformation could be accelerated by adding seeds for realizing the desired living supramolecular polymerization. Besides, this transformation of parallel/perpendicular aromatic stacking accompanied by time-dependent emission change from red to yellow is employed to dynamic cell imaging, largely avoiding the background interferences. The coopetition relationship of different aromatic stacking for metastable supramolecular systems is expected to serve as an effective strategy towards pathway-controlled functional materials.

In biological systems, kinetically trapped supramolecular aggregation is of paramount importance in achieving complex biomolecules with extraordinary structures and functions, such as intracellular transports[1,2], protein folding[3,4], and spontaneous self-healing[5,6]. Inspired by this, chemists' research interests are recently shifting from thermodynamic process to the implementation of metastable supra-molecular polymerization under kinetic control in artificial scenario[7–13]. The most widespread focus of the design of supramolecular polymerization with pathway complexity[14,15] proposed by Meijer et al. is to involve hydrogen interactions into the monomeric structures, which allows the formation of metastable species trapped by intramolecular hydrogen bonds and, finally, thermodynamic stable ones driven by intermolecular ones[16–25]. Apart from these intra-/inter-molecular hydrogen interactions widely reported by Würthner, Sán-chez, Fernández et al.[19–25], the combination of hydrogen/halogen interactions[26,27], hydrogen/π–π interactions[28–31], donor/acceptor

[1]School of Chemistry and Chemical Engineering, Northwestern Polytechnical University, Xi'an 710072, China. [2]Division of Chemistry and Biological Chemistry, School of Physical and Mathematical Sciences, Nanyang Technological University, Singapore 637371, Singapore. [3]These authors contributed equally: Zhao Gao, Xuxu Xie. ✉e-mail: happytw_3000@nwpu.edu.cn

**Fig. 1 | Chemical structures of monomers and energy landscapes of supramolecular polymerization. a** Chemical structures of the targeted monomers **1–2** and the control monomers **3–5**. **b** Schematic representation for the metastable supramolecular polymerization enabled by coopetition between parallel and perpendicular aromatic stacking. The metastable aggregates are trapped by face-to-face stacking (green background), while the thermodynamically stable one is driven by edge-to-face (red background) and offset (blue background) stacking. The edge-to-face stacking of indoles results in multifold C-H···π interactions. Red arrow represents the slow time evolution from 0D nanoparticles to 2D nanosheets. The blue arrow indicates the fast-living supramolecular polymerization process upon adding seeds.

interactions[32,33], metal/hydrophobic interactions[34–36], and *trans/cis* photoisomerization[37–40] are exploited to selectively steer the formation of kinetically or thermodynamically aggregates. Despite these advances in metastable supramolecular polymerization, sophisticated molecular structures possessing delicate relationships of cooperative, competitive, or orthogonal interactions were regarded as the necessities for achieving pathway control[41]. Thus, for metastable supramolecular polymeric system to become a general and versatile synthetic method, a minimalistic molecular model featuring accessible noncovalent driving modes are highly desirable to design that should be easy to control and predict the distinct outcomes derived from identical initial state.

To attain this goal, a feasible strategy is to integrate adjustable aromatic rings into one monomer. Different from other single noncovalent stacking modes, aromatics mainly interact with each other in parallel (offset and face-to-face stacking) and perpendicular (edge-to-face stacking) ways[42,43]. According to Hunter–Sanders electrostatic model[44], only offset and edge-to-face stacks are energy minima, while face-to-face stacking is rare because of the intrinsic thermodynamic unfavorable nature caused by electrostatic repulsion. In spite of this, previous works have constructed stable face-to-face aggregates through hydrogen bonds auxiliary anchoring[45–48], electrostatic interactions induced assembly[49–51], and confinement effect[52–54]. We have recently employed face-to-face aromatic cation-π forces to obtain dimensional polymorphic supramolecular polymers[55–59]. These reports are mainly focused on the aromatic interactions induced by thermodynamic products. With respect to the kinetically controlled aggregation mediated by these three aromatic stacking modes, it remains elusive and needs to be addressed. Here, we envisioned that if a π-conjugated monomer could first be competitively trapped in the manner of face-to-face stacking without any other synergetic interactions, it would spontaneously slide to thermodynamically favoured species under the cooperative driving of the two other more stable stacking modes. This cooperation coupled with competition (i.e., coopetition) driven relationship of parallel/perpendicular aromatic stacking is rare and expected to serve as a simple yet versatile solution for regulating the energy landscapes of metastable aggregation accompanied by variable optical signals under an extensive solvent and concentration conditions.

As a proof of concept, we proposed a coopetition-driven metastability strategy for parallel/perpendicular aromatic stacking to construct metastable supramolecular polymers. A series of π-conjugated monomers have been designed and synthesized (Fig. 1a). For simplicity and clarity, we only preserved the possible edge-to-face stacking of lateral indoles, as well as the offset and face-to-face stacking position on aromatic core. Remarkably, for monomer **1** with a sizeable anthryl core, it follows a metastable supramolecular polymerization (Fig. 1b). The kinetically trapped zero-dimensional (0D) nanoparticles slowly transform to the thermodynamically favored two-dimensional (2D) nanosheets. Even though for unilateral **2**, the kinetic behavior is also observed. In contrast, **3–5** undergo commonly thermodynamic supramolecular polymerization process. We have carefully investigated the differentiation in these supramolecular polymeric processes and given a plausible explanation. The face-to-face stacking of anthryl core in **1–2** is firstly trapped as metastable species by virtue of the competitive driving with the other two stacking modes, which spontaneously slipped into the most stable state driven by the cooperation of edge-to-face stacking of indoles and offset stacking of anthryl. This coopetition driving of aromatic stacking could be well modulated by varying the aromatic size and steric hindrance. When the aromatic core is as small as phenyl and naphthyl, **3–4** are mainly driven by edge-to-face stacking of indoles because the competitive face-to-face stacking could be ignored. Upon introducing two tert-butyls on anthryl core, the face-to-face stacking of **5** is immobilized. Based on the kinetic-to-thermodynamic transformation, **1** could be used for seed-triggered living supramolecular polymerization and dynamic cell imaging.

## Results

### Metastable supramolecular polymerization of 1

The supramolecular polymerization behavior of the targeted monomer **1** was first investigated. We recorded UV–Vis absorption and fluorescence spectra in a wide variety of solvents to understand the possible aggregation of **1**. The absorption spectra of **1** in the good solvent of N, N-dimethylformamide (DMF, $c = 3.00 \times 10^{-5}$ M) showed strong bands centered at 472 nm (Supplementary Fig. 1), belonging to the π–π* transition of the anthryl core[60]. Switching the solvent to $H_2O$/DMF (24:1, $v/v$), the absorption band blue-shifted to 463 nm (Supplementary Fig. 1), while a Tyndall effect occurred (Supplementary Fig. 2). Meantime, the fluorescence band of **1** centered at 503 nm showed obvious red-shift to 615 nm (Supplementary Fig. 3). The solution color varied from green-yellow to yellow, and the emission color changed

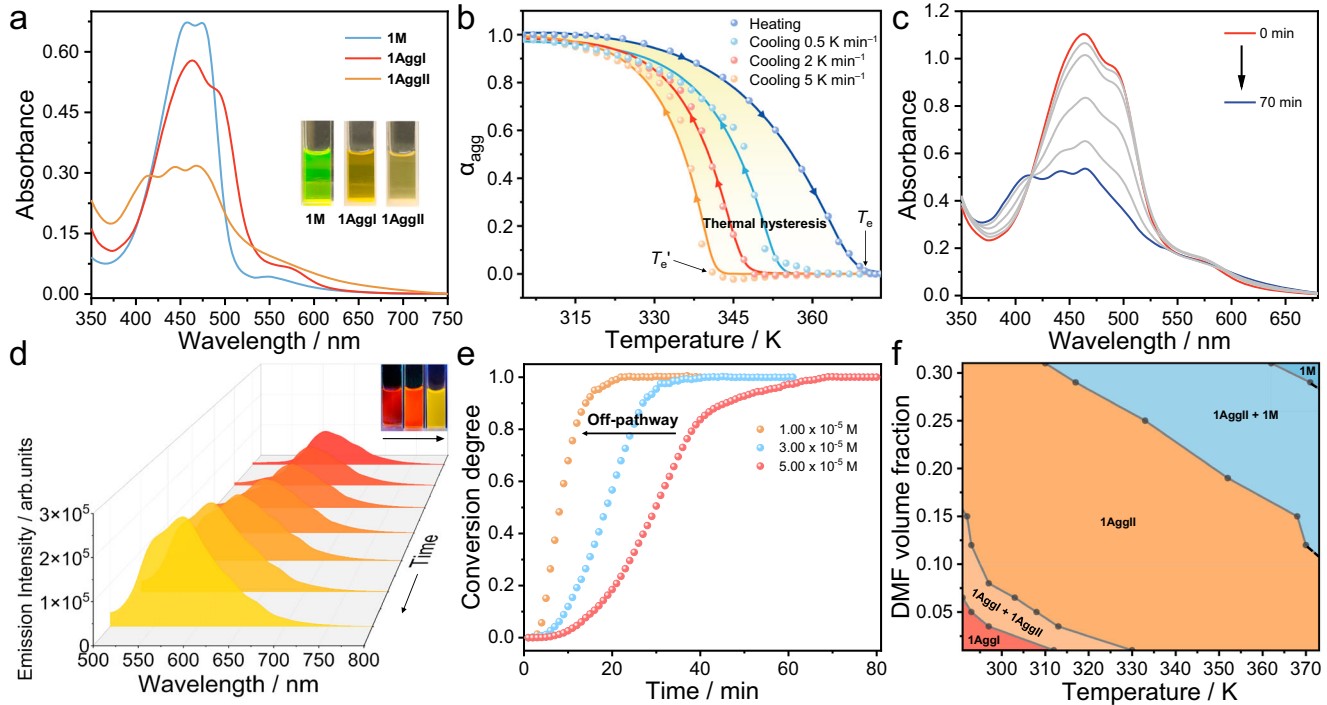

**Fig. 2 | Studies of the metastable supramolecular polymerization of 1. a** UV–Vis absorption spectra of **1** in H$_2$O/DMF (24:1, $v/v$, $c = 3.00 \times 10^{-5}$ M) at 370 K (blue line, **1 M**), fast cooling to 298 K (red line, **1AggI**), and slowly cooling to 298 K (orange line, **1AggII**). Inset: photographs of **1 M, 1AggI** and **1AggII** under natural light. **b** Temperature-dependent $\alpha_{agg}$ of **1AggII** calculated from the absorption intensities at $\lambda = 475$ nm observed in the heating and cooling processes at the rate of 0.5, 2 and 5 K min$^{-1}$. The yellow area represents the thermal hysteresis range for kinetically stabilizing **1 M**. Time-dependent **c** absorption and **d** fluorescence spectral variations of **1AggI** in H$_2$O/DMF (24:1, $v/v$, $c = 5.00 \times 10^{-5}$ M) at 298 K. Inset of **d**: photographs of **1AggI** aging for 0, 45 and 90 min under UV light. $\lambda_{ex} = 365$ nm. **e** Kinetic profiles of the conversion from **1AggI** to **1AggII** at the concentrations of $1.00 \times 10^{-5}$ M, $3.00 \times 10^{-5}$ M and $5.00 \times 10^{-5}$ M. The arrow shows the shortened lag time of the conversion upon decreasing concentration. **f** State diagram of the predominance of the respective species of **1** in dependency of temperature and DMF volume fraction in H$_2$O/DMF mixture.

from green to red (Supplementary Figs. 2 and 3). Such phenomena demonstrate the transition from the monomeric state (named as **1 M**) to the supramolecular aggregated state of **1** (named as **1AggI**).

Intriguingly, upon slowly cooling the solution of **1** at a rate of 0.5 K min$^{-1}$, the resultant products possess distinct optical properties compared to **1AggI**, which exhibited a triple absorption band with $\lambda_{max}$ at 470 nm and a yellow emission band centered at 583 nm (Fig. 2a, orange line, and Supplementary Fig. 4). The phenomena indicate that a different type of supramolecular polymerization product is formed (named as **1AggII**). The temperature-dependent aggregated degree ($\alpha_{agg}$) of **1AggII** estimated from the absorption band at $\lambda = 475$ nm observed in the cooling process exhibited a non-sigmoidal curve (Fig. 2b, orange dots), indicating a cooperative nucleation-elongation supramolecular polymerization process from monomer **1** to **1AggII**. Upon heating the solution of **1AggII**, the heating curve (Fig. 2b, blue dots, and Supplementary Fig. 5) was not overlapped with the cooling curve, that is, a pronounced hysteresis loop appeared (Fig. 2b, yellow background area). The curves fitted well with the mass balance model[61], acquiring the clearly distinguished critical elongation temperature $T_e'$ and $T_e$ as 341.3 K and 370.1 K, respectively. Moreover, $T_e'$ value was shifted to a higher temperature as the cooling rate slowed from 5 to 0.5 K min$^{-1}$ (Fig. 2b, orange and cyan dots). These results unambiguously verify that the supramolecular polymerization of **1** is not able to follow the rapid temperature change and should undergo the metastable pathway under kinetic control.

It is well-known that the kinetically metastable species will spontaneously transform into the thermodynamically stable one over time. To get insight into this evolution, time-dependent absorption and fluorescence spectra variation were employed. Upon standing **1AggI** at 298 K for a period of time, the absorption band of **1AggI** gradually

transformed into that of **1AggII** (Fig. 2c). In the meantime, the fluorescence band of **1AggI** blue-shifted from $\lambda_{max}$ at 615 nm (absolute fluorescence quantum yield, $\Phi_F = 1.63\%$, Supplementary Fig. 6) to 583 nm ($\Phi_F = 4.13\%$, Supplementary Fig. 7), with an emission changes from deep red to yellow (Fig. 2d and inset). Therefore, **1AggI** is assigned as the metastable species, while **1AggII** is the thermodynamically favored one. By monitoring the absorption spectral change from **1AggI** to **1AggII** at 463 nm overtime at $5.00 \times 10^{-5}$ M, a sigmodal curve with a lag time of ~12.5 min was obtained (Fig. 2e, red dot, and Supplementary Fig. 8). Such phenomenon is ascribed to an autocatalytic growth of monomers that consists of nucleation and elongation processes. The spontaneous nucleation of **1 M** from **1AggI** was kinetically trapped longer with decreasing temperature (Supplementary Fig. 9). Importantly, the nucleation rate constant ($k_n$) was determined to be $1.17 \times 10^{-3}$ min$^{-1}$, which is increased upon decreasing concentration (Fig. 2e and Supplementary Fig. 10). The corresponding lag time shortened to 2 min, reflective of the accelerated transition. These concentration dependence results indicate that the transformation from **1AggI** to **1AggII** undergoes an off-pathway process[20], which requires the disassembly of **1AggI** into the monomer **1 M** to further aggregate into the thermodynamically stable product **1AggII**. Apart from the good solvent of DMF, we found that isopropanol, methanol, acetone, tetrahydrofuran, and dimethylsulfoxide (DMSO) were also suitable for the kinetic assembly of **1** (Supplementary Fig. 11).

Considering that the aggregated state of **1** is closely correlated to temperature and solvent, we then sought to draw the comprehensive supramolecular polymeric state diagrams[62]. Temperature-dependent absorption spectra of **1AggII** ($c = 2.00 \times 10^{-5}$ M) were separately monitored at 463 nm in H$_2$O/DMF mixtures with different DMF volume fraction ranging from 1% to 31% (Supplementary Fig. 12 and

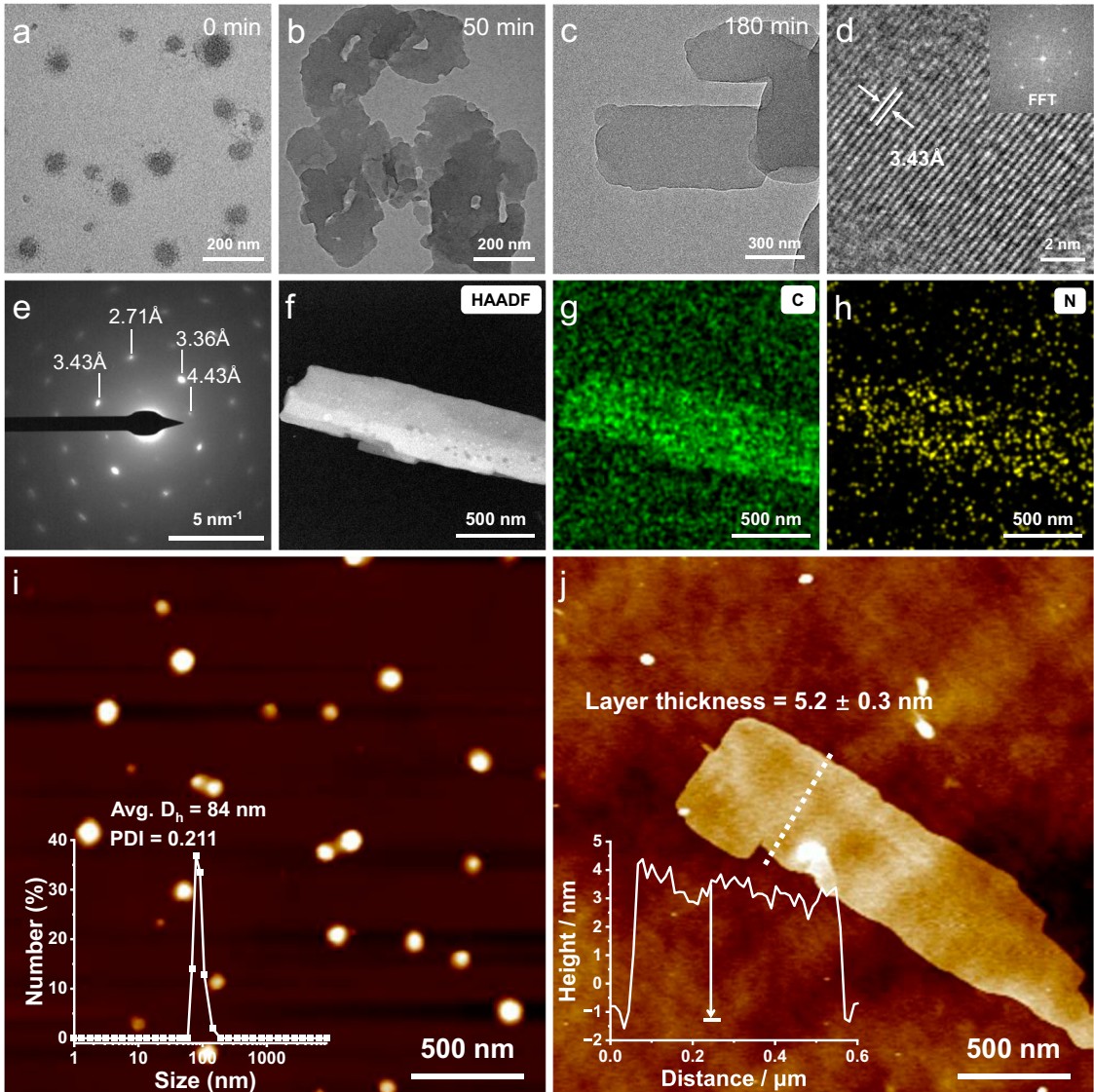

**Fig. 3 | Characterization of the morphological evolution of 1.** TEM images of **a** the metastable species **1AggI** and after aging for **b** 50 min and **c** 180 min to obtain the thermodynamically stable species **1AggII**. **d** High-resolution TEM image showing the lattice fringes and **e** the corresponding SAED pattern of **1AggII**. **f–h** HAADF-STEM image and the corresponding EDS for elemental mapping of **1AggII**. AFM images of **i 1AggI** and **j 1AggII**. Inset of **i**: DLS results of **1AggI**. Inset of **j**: the height profile of **1AggII**. The height was the average value along the white dash line. The samples were obtained by drop-casting the $H_2O$/DMF (24:1, $v/v$, $c = 3.00 \times 10^{-5}$ M) solution of **1** on copper grid for TEM experiments and on silicon wafer for AFM experiments. Experiments were repeated at least three times with similar results.

Supplementary Table 5). The temperatures for the transformation of **1AggI**-to-**1AggII**-to-**1M** were determined and plotted as a function of the solvent composition, displaying the precise state diagram (Fig. 2f). The boundary between adjacent states is not sharp (the mixture states of **1AggI**+**1AggII** and **1AggII**+**1M**), ascribing to the consecutive transformation. It should be noted that **1AggII** existed in a relatively broad temperature range of 312 to 373 K and broad DMF volume fraction of 6.5% to 31%, probably resulting from its thermodynamically stable feature. This temperature- and solvent-dependent state diagram would provide guidance for controlling over the aggregated state of **1** toward further applications.

The morphology evolution of **1** was then evaluated. Transmission electron microscopy (TEM) image showed that nanoparticles were formed for metastable **1AggI** in the initial stage (0 min, Fig. 3a). Subsequently, the nanoparticles began to fuse spontaneously (Fig. 3b), and the regular 2D nanosheets of **1AggII** were eventually formed after 180 min (Fig. 3c). The high-resolution TEM image exhibited clearly visible fringes with a lattice parameter of 3.43 Å (Fig. 3d), while the

corresponding selected-area electron diffraction (SAED) pattern showed clearly diffraction spots with distances ranging from 2.71 to 4.43 Å (Fig. 3e). These are the effective aromatic stacking distance for lamellate structures. The 2D nanosheets were further verified by the high-angle annular dark-field scanning transmission electron microscopy (HAADF-STEM) and energy dispersive spectroscopy (EDS). The EDS mapping showed that the typical elements of C and N were uniformly distributed throughout the sample areas (Fig. 3f–h and Supplementary Fig. 13), signaling the composition of monomer **1**. The quantitative structure parameters of **1AggI** and **1AggII** were then obtained by atomic force microscopy (AFM) and dynamic light scattering (DLS) measurements (Fig. 3i–j). Nanoparticles of ~84 nm DLS hydrodynamic diameters with polydispersity index (PDI) of 0.211 for **1AggI** and the average thickness of the lamella of $5.2 \pm 0.3$ nm for **1AggII** were observed separately. Overall, the above experimental results unambiguously verified that the elaborate monomer **1** aggregated in a metastable supramolecular polymerization manner to form the well-regulated 2D nanosheets.

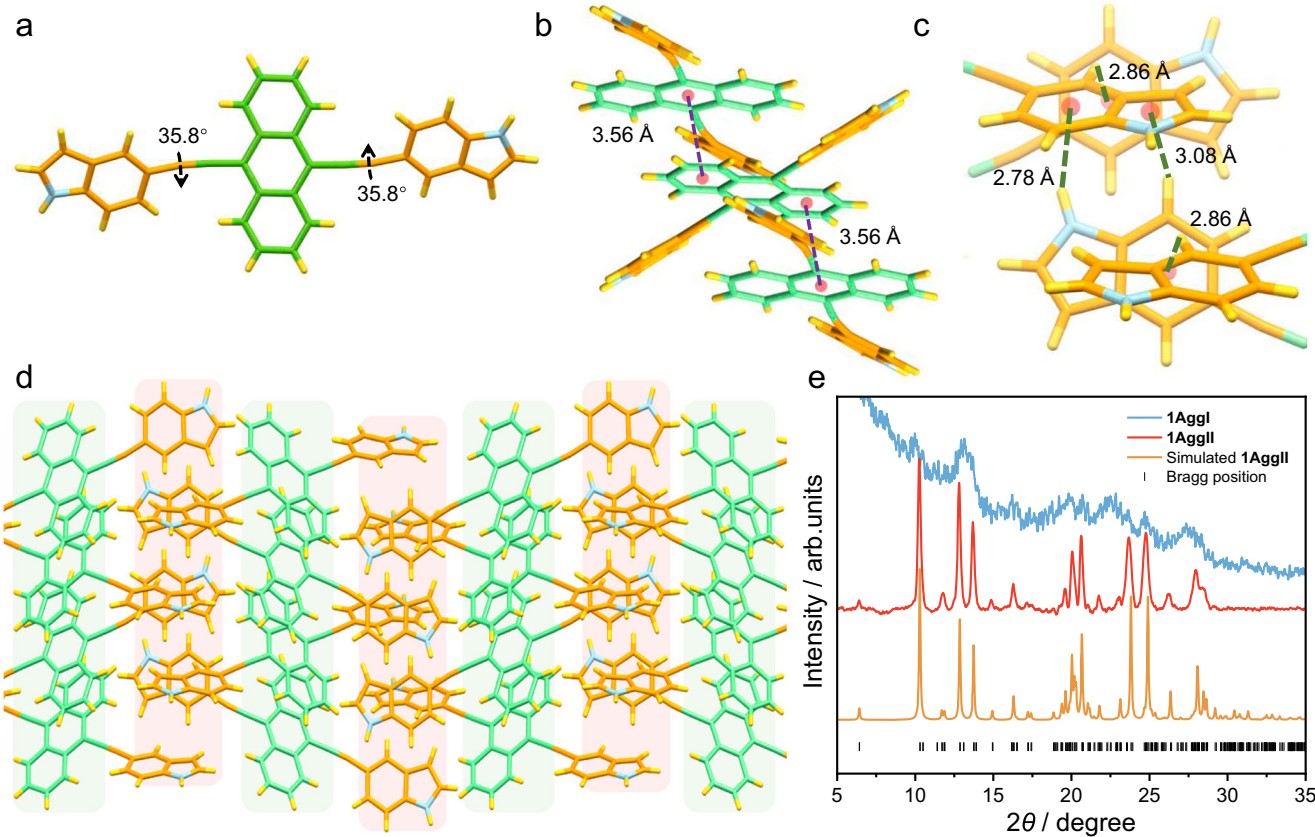

**Fig. 4 | Molecular stacking mode of the thermodynamically stable 1AggII.**
**a**–**d** X-ray single-crystal structures of **1AggII** (CCDC: 2351020) at different views. The orange and green colored background in **d** indicate the position of edge-to-face stacking of indoles and offset stacking of anthryl, respectively. **e** The experimental and calculated PXRD pattern of **1AggI** and **1AggII**.

We further studied the supramolecular polymerization behaviors of analog monomer **2**, which features only one side of indoles and the other side replaced with phenyl (Fig. 1a). Upon fast cooling or freshly preparing the H₂O/DMF (3:1, $v/v$, $c = 3.00 \times 10^{-5}$ M) solution of **2**, the kinetically metastable species (**2AggI**) were obtained (Supplementary Fig. 14). The thermodynamically stable one (**2AggII**) obeying cooperative supramolecular polymerization mechanism was formed by slowly cooling (1 K min⁻¹) or standing **2AggI** at room temperature for 90 min (Supplementary Fig. 15). Moreover, an interesting hysteresis loop between the heating-cooling cycle was observed (Supplementary Fig. 16), which represented that **2 M** was kinetically trapped and inactivated within the temperature range between $T_e$ (361.8 K) and $T_e'$ (346.2 K). Similar to **1AggI**, **2AggI** was regarded as an off-pathway product, as reflected by the accelerated kinetic-to-thermodynamic transformation when increasing concentration (Supplementary Fig. 17). It is worth noting that the spectral shape and wavelength of **2** is familiar with that of **1**, characteristic of the analogous aggregation mode. This conclusion was also supported by the similar 2D morphologies in the TEM, SAED and DLS experiments (Supplementary Fig. 18).

As a comparison, the supramolecular polymerization behaviors of the control monomer **4** was then investigated. When monitoring $\alpha_{agg}$ value at $\lambda_{max} = 338$ nm versus temperature, a sigmoidal melting curve was observed (Supplementary Fig. 19), indicating the involvement of an isodesmic supramolecular polymerization mechanism[63]. Nanoparticles of **4** with ~95 nm diameters were formed based on the morphologies and DLS measurements (Supplementary Fig. 20). Differentially, for monomer **3**, two thermodynamically stable products were separately obtained. One type of the aggregated nanoparticles **3** (named as **3AggI**) was first captured by a freshly preparing sample in the solvent of H₂O/DMF (3:1, $v/v$), which is very stable upon long-time standing at room temperature (Supplementary Fig. 21) and conforms to the isodesmic mechanism (Supplementary Fig. 22). After heating-cooling cycle, the larger-sized aggregate (named as **3AggII**) was formed (Supplementary Fig. 23). **3AggII** was also the thermodynamic species (Supplementary Fig. 24), confirming by the overlapped sigmoidal heating and cooling curves (Supplementary Fig. 25).

## Coopetition-driven metastability mechanism

After elucidating the metastable supramolecular polymerization pathway of **1**, we further confirmed the molecular stacking mode of the two distinct aggregates. As the thermodynamically stable property of **1AggII**, we successfully obtained its single crystal by allowing a methanol solution to evaporate slowly over 6–7 days. The rigid monomer **1** crystallized in the monoclinic P2₁/n space group, in which the bilateral indole units are parallel to each other and showed torsional angles of 35.8° to the anthryl core (Fig. 4a). The central ring of anthryl was arranged parallelly with the marginal ring of the adjacent anthryl, showing a poor offset stacking with a distance of 3.56 Å (Fig. 4b). Interestingly, multifold C-H⋯π interactions resulted from the edge-to-face stacking of indoles were observed, which connect monomers along the $a$-axis with the distances ranging from 2.78 to 3.08 Å (Fig. 4c). Governed by these interactions, monomer **1** mainly aligning in-plane extension resulted in an ordered 2D structure (Fig. 4d). The structure of **1AggII** was then evaluated by powder X-ray diffraction (PXRD) analysis. In order to maintain the same stacking mode as that measured in spectroscopic experiments, the sample was prepared by fast freeze-drying the solvent of the solution of **1AggII**.

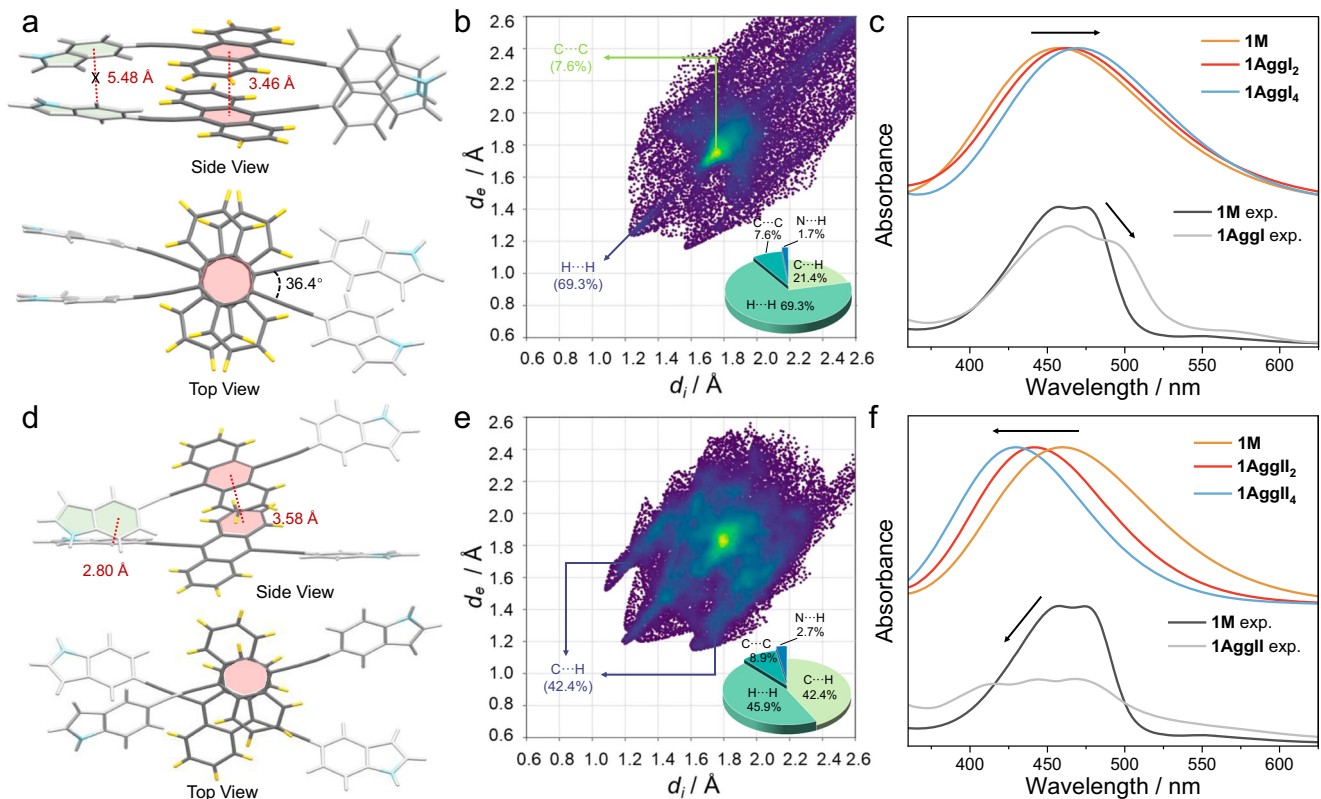

**Fig. 5 | Confirmation of the molecular packing of 1AggI by spectroscopic results and theoretical analysis.** The optimized dimer structures of **a 1AggI** and **d 1AggII** at the side view and top view, together with the corresponding calculated and experimental electronic transition spectra of **c 1AggI** (including monomer **1 M**, dimer **1AggI₂**, and tetramer **1AggI₄**) and **f 1AggII** (including monomer **1 M**, dimer **1AggII₂**, and tetramer **1AggII₄**) based on the DFT calculations by a B3LYP/6-31 G (d) basis set. The arrows in **c** and **f** represent the spectral variation trends. Fingerprint graphics based on Hirshfeld surfaces for non-covalent contact distributions including C···H, C···C, H···H, and N···H of **b 1AggI₂** and **e 1AggII₂**, along with the corresponding pie charts (insets).

The intense diffraction peaks with narrow half-peak widths showed an excellent correlation with the simulated result from the single-crystal structure (Fig. 4e, red and orange lines), characteristic of the identical aggregation form. Moreover, the distances measured from the diffraction peaks were located at 19.6–29.2° (4.52–3.06 Å), which are in good agreement with those measured in the SAED pattern (Fig. 3e). Comparatively, a different PXRD pattern with broader and shifted diffraction peaks was obtained for **1AggI** (Fig. 4e, blue line), indicating the distinct aggregation with relatively lower crystallinity.

To get further insights into the stacking mode of **1AggI**, spectroscopic experiments and quantum chemical calculations were employed. Specifically, geometries of monomers, dimers (**1AggI₂**), and tetramers (**1AggI₄**) of **1** were built and optimized via density functional theory (DFT) calculations by a B3LYP/6-31 G(d) basis set, and the corresponding electronic transition spectra were simulated by time-dependent density functional theory (TD-DFT) calculations at the same computational level. The optimized metastable **1AggI₂** and **1AggI₄** exhibited a face-to-face stacking, in which the central ring of anthryl core was parallel to the adjacent anthryl central ring (Fig. 5a and Supplementary Fig. 26). The π–π distance was calculated to be 3.46 Å, and the rotation angle of anthryl was as small as 36.4°. However, even though the indoles stacking existed, the distance of C-H and phenyl were far apart from each other (5.48 Å), thus precluding the possibility of C-H···π interactions. Such results implied that the face-to-face stacking of anthryl cores dominates the metastable **1AggI**. Hirshfeld surfaces further confirmed the crucial role of this π–π interactions. The intermolecular contact distributions based on the Hirshfeld surfaces were highlighted on the fingerprint graphics (Fig. 5b and Supplementary Fig. 27). H···H and C···C occupied the largest fraction of 69.3% and 7.6%, respectively, suggesting the mainly driving forces of π–π interactions. To verify the rationality of the optimized structure of **1AggI**, the resulted spectroscopic signatures were compared. The calculated electronic transition spectra of **1AggI₂** and **1AggI₄** displayed a sequential red-shift with respect to the monomer absorption, which is consistent with the experimentally measured trends (**1 M** exp. versus **1AggI** exp., Fig. 5c). The spectrum of **1AggI₄** overlapped well with **1AggI** exp. (Supplementary Fig. 28). Additionally, for **1AggII**, DFT calculations (the same computational basis set of **1AggI**) for its dimers and tetramers successfully reproduced both aggregations compared to the single-crystal results (Fig. 5d and Supplementary Fig. 29). The fraction of C···H increased up to 42.4% in the fingerprint graphics, reflective of the domination of C-H···π interactions (Fig. 5e and Supplementary Fig. 30). Moreover, the direction of spectral shift was opposite to that of **1AggI**, agreeing well with the measured results (Fig. 5f). Based on the above-proposed stacking mode of **1AggI** and **1AggII**, $\Delta G$ (Gibbs free energy) of the supramolecular polymerization process of **1AggII** was calculated to be −2835.11481348 Hartree, which is more stable in energy than the kinetic metastable **1AggI** (−2835.06764086 Hartree).

We further performed $^1$H NMR measurements to illustrate the difference in non-covalent interactions in terms of the molecular aggregation of **1AggI** and **1AggII**. In order to probe the clear signals, the mixed solvent D₂O/DMSO-$d_6$ (3:7, $v/v$) was chosen, which agreed to obtain the identical metastable supramolecular polymerization results as before (Supplementary Fig. 31). The well-defined sharp signals of aromatic protons on the freshly prepared **1AggI** gradually shifted upfield, broadened and submerged in the baseline (Fig. 6a). This strong shielding effect signified the transformation from small aggregates to large-sized supramolecular polymers. The protons correlation was then verified by 2D $^1$H − $^1$H nuclear Overhauser effect spectroscopy

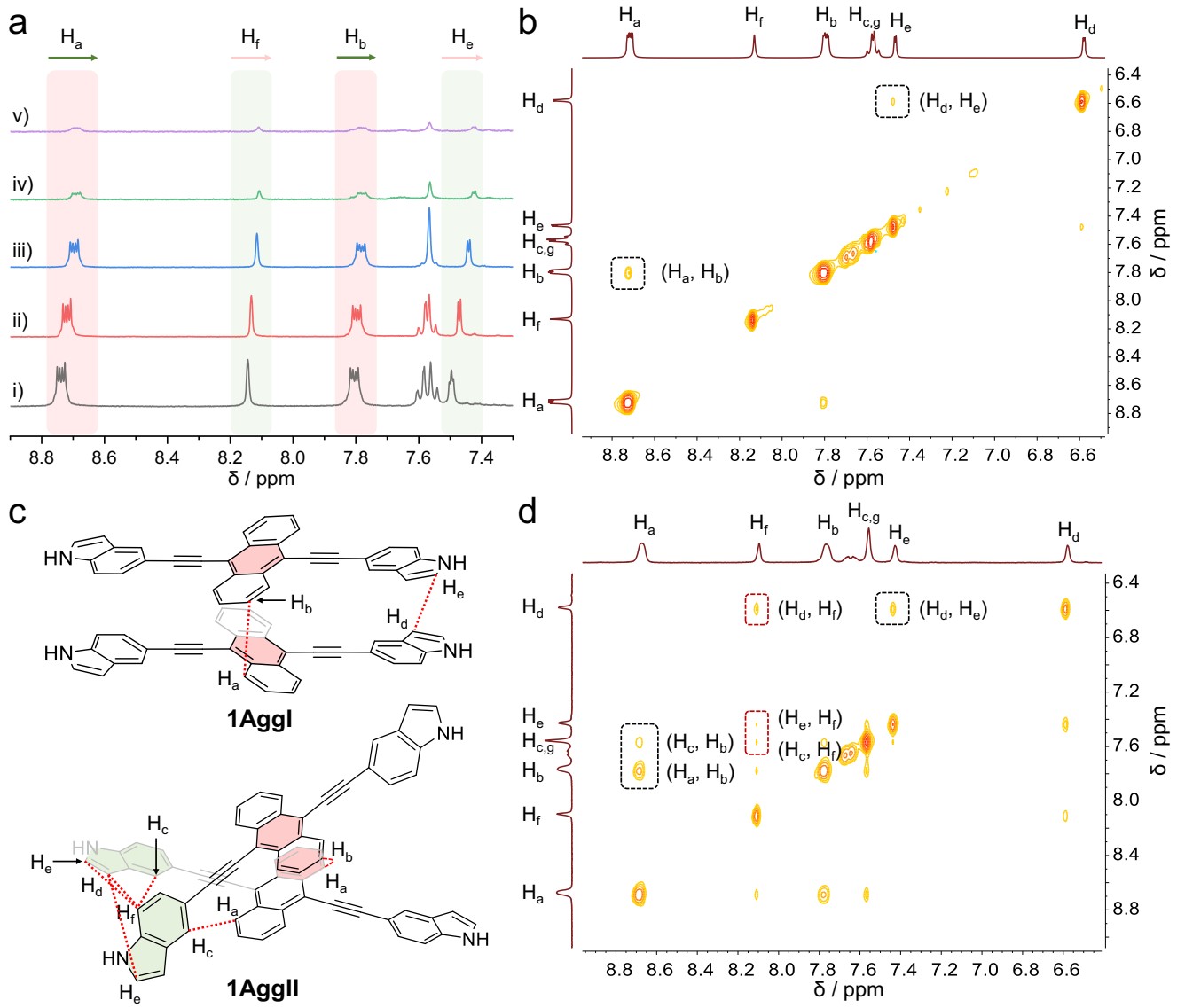

**Fig. 6 | Analysis of the non-covalent interactions for 1AggI and 1AggII by ¹H NMR measurements. a** Time-dependent partial ¹H NMR (400 MHz, 298 K, 5.00 mM, D₂O/DMSO-$d_6$, 3:7, $v/v$) spectral variations of **1AggI**: (i) 0 min, (ii) 10 min, (iii) 20 min, (iv) 30, and (v) 40 min. Partial ¹H-¹H NOESY NMR (400 MHz, 298 K, 5.00 mM, D₂O/DMSO-$d_6$, 3:7, $v/v$) spectra of **b 1AggI** and **d 1AggII**, and **c** the corresponding correlated protons positions of each dimers.

(NOESY) experiments. Apparent NOE correlated signals were observed between the 1-position H$_a$ and 2-position H$_b$ on the marginal anthryl ring (Fig. 6b, c), reflective of a strong π–π interaction of anthryl in a face-to-face manner. Besides, the neighbouring indoles exhibited a weak correlation of aromatic protons H$_d$ and H$_e$, belonging to the π-π interaction. These results are very consistent with the molecular stacking model of **1AggI** (Fig. 5a). In comparison, apart from the similar correlations of (H$_a$, H$_b$) and (H$_d$, H$_e$), the NOE signals of (H$_d$, H$_f$), (H$_c$, H$_f$) and (H$_e$, H$_f$) representing multifold C-H···π interactions in indole units appeared for **1AggII** (Fig. 6c, d). Thus, the combined results from all the above single-crystal structures, spectroscopic results, theoretical analysis, and ¹H NMR spectra agree with a parallel face-to-face arrangement of anthryl for **1AggI**, whilst a repetitive unit for **1AggII** drove by C-H···π interactions of indoles along with weak offset stacking of anthryl.

A series of control experiments were then performed to investigate the coopetition relationship of parallel stacking of anthryl and perpendicular C-H···π interactions of indoles for the metastable supramolecular polymerization. As discussed before, monomer **2** also underwent metastable supramolecular polymerization process.

Differently, metastable **2AggI** lasted longer time than that of **1AggI** (lag time: 9 min versus 2 min, $k_n$: $3.49 \times 10^{-3}$ min$^{-1}$ versus $1.16 \times 10^{-2}$ min$^{-1}$, Supplementary Figs. 10 and 32). We rationalized that the more stable feature of **2AggI** is due to the weakened competitiveness of C-H···π interactions in **2AggII** with unilateral indoles than in **1AggII** with bilateral indoles. This conclusion was reflected by the single-crystal structure of **2AggII** (Supplementary Fig. 33). Unilateral C-H···π interactions between indoles connected monomers along the *a*-axis with distances ranging from 2.70 to 2.85 Å. The other side phenyl showed very weak interactions, negligible contributing to the aggregation of **2**. The C···H distribution in the fingerprint graphics was calculated to be 46.9% for **2AggII** (Supplementary Fig. 34), which is more than that of **2AggI** (19.4%, Supplementary Fig. 35). Combination with the analyses of aggregation state diagram, ¹H–¹H NOESY, DFT calculations and simulated absorption spectra (Supplementary Figs. 36–38 and Supplementary Table 6), it is concluded that the coopetition-driven metastability strategy of parallel/perpendicular aromatic stacking is applicable to other analogous monomers.

Besides, for monomer 9,10-bis(phenylethynyl)anthracene, replacing bilateral indoles with phenyl groups, the absorption band at

450–550 nm belonging to anthryl remained unchanged upon long-term standing at room temperature (Supplementary Fig. 39), manifesting its thermodynamically stable character. These results evidenced that the parallel and perpendicular aromatic interactions are indispensable for developing metastable supramolecular polymers. To prevent the spontaneously transformation of metastable **1AggI**, we introduced two tert-butyl groups at the 2- and 6-position of anthryl to obtain the elaborate monomer **5** (Fig. 1a). The absorption and fluorescence spectra of the fresh prepared H₂O/DMF (3:1, $v/v$, $c = 3.00 \times 10^{-5}$ M) solution of **5** (named as **5AggI**) sparingly changed even after one week (Supplementary Fig. 40), indicating the good stability. The spectral shape and wavelength of **5AggI** is familiar with those of **1AggI** and **2AggI**, reflective of the immobilized face-to-face stacking mode. This is ascribed to the self-locking of **5**-dimer benefiting from the steric hindrance effect (Supplementary Fig. 41)[64,65]. Upon heating **5AggI** to monomer then cooling to room temperature, an unchanged spectral signal was observed (assigning to **5AggII**, Supplementary Fig. 42), which is also the thermodynamic state.

Thus, taking into account all these supramolecular polymerization behaviors of monomers **1–5**, we concluded that the coopetition of parallel stacking of anthryl and perpendicular C-H···π interactions of indoles mediates the supramolecular polymerization pathway evolutions. Initially, due to the stacking could be ignored when the aromatic core is as small as phenyl, **4** is mainly driven by C-H···π interactions resulted from the edge-to-face stacking of indoles (Supplementary Fig. 43). Furthermore, the stacking could no longer be neglected upon enlarging aromatic core to naphthyl (situation for **3**). Even so, the multifold C-H···π interactions still dominate, resulting in the two stable supramolecular polymers driving by edge-to-face stacking of indoles assisted with offset stacking of naphthyl (Supplementary Figs. 44 and 45). Finally, when the aromatic core expands to anthryl (situations for **1–2**), the kinetically trapped state is competitively formed via a face-to-face stacking of anthryl, then the spontaneously kinetic-to-thermodynamic transformation is occurred driven by cooperative of edge-to-face stacking of indoles and offset stacking of anthryl core.

## Seed-triggered living supramolecular polymerization

After understanding that the kinetic-to-thermodynamic transformation must cross the high energy barrier (Fig. 1b, red arrow), we proceeded to explore whether this barrier can be bypassed by adding seeds to achieve the living supramolecular polymerization (Fig. 1b, blue arrow). This controlled polymerization method is regarded as a promising way to mimic nature for creating sophisticated supramolecular systems with various compositions, topologies and functions[7,8]. By sonicating the solution of **1AggII** or **2AggII** for 5 min at 298 K, the corresponding seeds were prepared (**1seed** and **2seed**). Although **1AggII** and **2AggII** were split into small pieces, the dimensionalities and the optical properties of the resulting **1seed** and **2seed** retained the same (Supplementary Fig. 46).

Upon adding 1 mol% of **2seed** into the solution of **2AggI** at 296 K, the accelerated transformation from **2AggI** to **2AggII** with eliminated lag time was achieved by monitoring the absorbance band variation at 450 nm versus time (Fig. 7a). The total transformation time remarkably shortened from 100 min to 2.2 min, which could be adjusted by varying seed concentrations. The logarithm of the transformation rate is directly proportional to the logarithmic concentration of **2seed** (Fig. 7b), which is regarded as a favorable chain-growth polymerization mechanism. To evaluate the seed cycles ability for the living growth of supramolecular polymers, the manifold cycles experiments as illustrated in the protocol were applied (Fig. 7c). Equal equivalent of **2seed** was first added into **2AggI**. After the transformation was completed, one equivalent of the polymerization product was removed to keep the overall amount of the sample constant, and then another one equivalent of **2AggI** was added for the subsequent polymerization

cycle. This procedure could be repeated for at least eight cycles (Fig. 7d), with a reduced polymerization rate by half for each cycle (Supplementary Fig. 47). AFM images of the polymerization products after each cycle exhibited a gradual increase in the area of nanosheets from $6.30 \times 10^{3}$ nm² (seeds) to $1.01 \times 10^{5}$ nm² (3$^{rd}$ cycle), $3.19 \times 10^{5}$ nm² (5th cycle), and 1.25 µm² (7th cycle) (Fig. 7e–h and Supplementary Fig. 48). Besides, the hydrodynamic diameters in DLS experiments also showed the growing trend in size from 122 nm of **2seed** to 1.99 µm of the last cycle samples (Supplementary Fig. 49). Likewise, **1seed**-triggered living supramolecular polymerization for the kinetically trapped **1AggI** was achieved (Supplementary Fig. 50). Moreover, hetero-seeds could also induced the living supramolecular polymerization[66]. Upon addition of **1seed** into **2AggI**, the transformation time from **2AggI** to **2AggII** was shortened within 4.5 min (Supplementary Fig. 51), reflective of the living character of this supramolecular polymerization.

## Dynamic cell imaging of the kinetic-to-thermodynamic transformation of 1

π-Conjugated fluorophores have been considered promising candidates in the biosensing and diagnosis fields[67,68]. However, the background interferences and possible artefacts limit the further development of these fields. Considering that **1** underwent a kinetic-to-thermodynamic transformation from the red emission **1AggI** to the yellow emission **1AggII**, we applied it as a potential dynamic fluorescent cell imaging agent, which is expected to better rule out artefacts and provide higher background contrast. To verify this hypothesis, we first performed the time-dependent fluorescence spectral variations of **1AggI** in the cell culture conditions. As shown in Fig. 8a, the intensity of the emission band of **1AggI** gradually enhanced and blue-shifted over time, which maintained the same variation trends as that measured in Fig. 2d. By monitoring the intensities at 550 and 600 nm, a sigmodal curve with lag time was separately observed (Fig. 8b). Therefore, the preliminary experiments verified the feasibility of dynamic cell imaging of **1**.

On this basis, A549 cells were selected as a model for the dynamic cell imaging experiments. We first evaluated the cytotoxicity of **1** by incubating with A549 cells for 24 h before measuring cell viability through an MTT assay. **1** showed negligible toxicity to A549 cells at a concentration of $0.80 \times 10^{-5}$ M (Supplementary Fig. 52). After staining A549 cells with **1** for 20 min, the indistinct fluorescence images in the red and green channels were immediately observed in the confocal laser scanning microscopy (CLSM) (Fig. 8c, f). Subsequently, the bright emission gradually emerged with increasing time (Fig. 8c–h), consistent with the spontaneous transformation kinetics associated with **1AggI** to **1AggII**. In sharp contrast, A549 cells stained with the control compound **5** exhibited invariable CLSM images (Supplementary Fig. 53), which is related to the immobilized thermodynamic **5AggI**. Dynamic cell imaging is also applicable to other cellular environments, such as B16-F10 cells (Supplementary Figs. 54 and 55). Thus, the dynamic cell imaging was successfully achieved by virtue of the spontaneous kinetic-to-thermodynamic transformation nature of **1** accompanied by fluorescence modulation.

## Discussion

In conclusion, we have successfully described a coopetition-driven strategy of parallel/perpendicular aromatic stacking for metastable supramolecular polymerization based on simple monomers consisting of lateral indoles and aromatic core. The kinetically trapped 0D nanoparticles of **1–2** were competitively obtained by the face-to-face stacking of anthryl core. Subsequently, by virtue of the cooperative edge-to-face stacking of indoles and offset stacking of anthryl, thermodynamically more stable 2D nanosheets were spontaneously formed. The coopetition relationship of different aromatic stacking was carefully discussed by single-crystal structures, ¹H NMR results, theoretical modeling, and control monomers **3–5**. Furthermore, the kinetic-to-thermodynamic transformation was exploited not only to

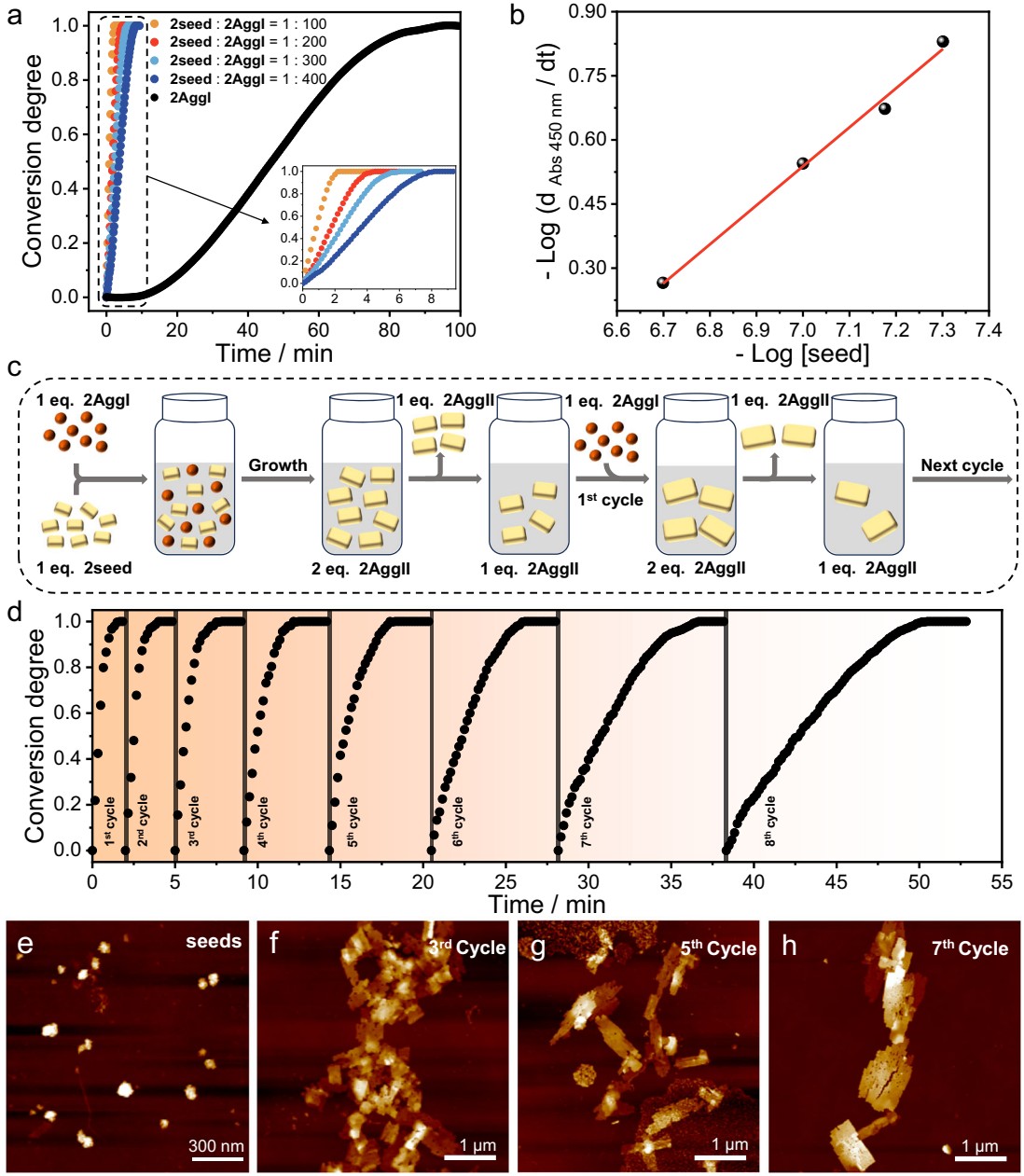

**Fig. 7 | Confirmation of the living supramolecular polymerization. a** Time course of seeded supramolecular polymerization of **2** monitored as changes in absorbance intensities at $\lambda = 450$ nm initiated by the addition of **2seed** to an $H_2O$/DMF (3:1, $v/v$, $c = 2.00 \times 10^{-5}$ M) solution of **2AggI** in the molar ratios of 1:100, 1:200, 1:300 and 1:400. Inset shows the partial amplified area. **b** Log-log plot of the rate of polymerization as a function of the concentration of **2seed**. The correlation coefficient of solid line linear fitting was 0.998. **c** Schematic representation of the experimental operation and mechanism of the seeds controlled living supramolecular polymerization cycles. eq. represents equivalent. **d** Time course of the absorbance intensities at $\lambda = 450$ nm during multicycle living supramolecular polymerization of **2**. The gray lines represent the time for removing equivalent of **2AggII** and adding equivalent of **2AggI**. AFM images of **e 2seed** and **f**–**h** the gradually enlarged nanosheets of **2AggII** obtained after the seeded supramolecular polymerization. The images were repeated at least three times with similar results.

the seed-triggered living supramolecular polymerization bypassing the high energy barrier, but also to the dynamic cell imaging without background interferences. We believe that the coopetition-driven metastability viewpoints can serve as a promising approach towards the synthetic methodology of metastable supramolecular systems and promote the development of new supramolecular functional materials.

## Methods
### Measurements

$^1$H NMR, $^{13}$C NMR, and $^1$H-$^1$H NOESY spectra were measured from Bruker Avance 400 instruments. High-resolution electrospray ionization mass spectra (HR-ESI-MS) were obtained on an LCMS-IT-TOF equipped with an ESI interface and ion trap analyzer. UV–Vis absorption spectra were performed on a Shimadzu UV-2600 spectrometer. Fluorescence spectra were recorded on a Hitachi FLS-4600 FL spectrophotometer. Quantum yield was measured by an Edinburgh FLS980 transient steady-state fluorescence spectrometer with an integrating sphere. TEM experiments were performed on a FEI Talos F200X TEM instrument. AFM images were obtained on a Bruker Dimension FastScan and Dimension Icon instrument. PXRD was recorded on a SHIMADZU XRD-7000 with a Cu Kα X-ray source ($\lambda = 1.540598$ Å). The single-crystal data were collected on Bruker D8 Venture. Cell imaging experiments were measured from a leica Stellaris 5 confocal laser fluorescence microscope.

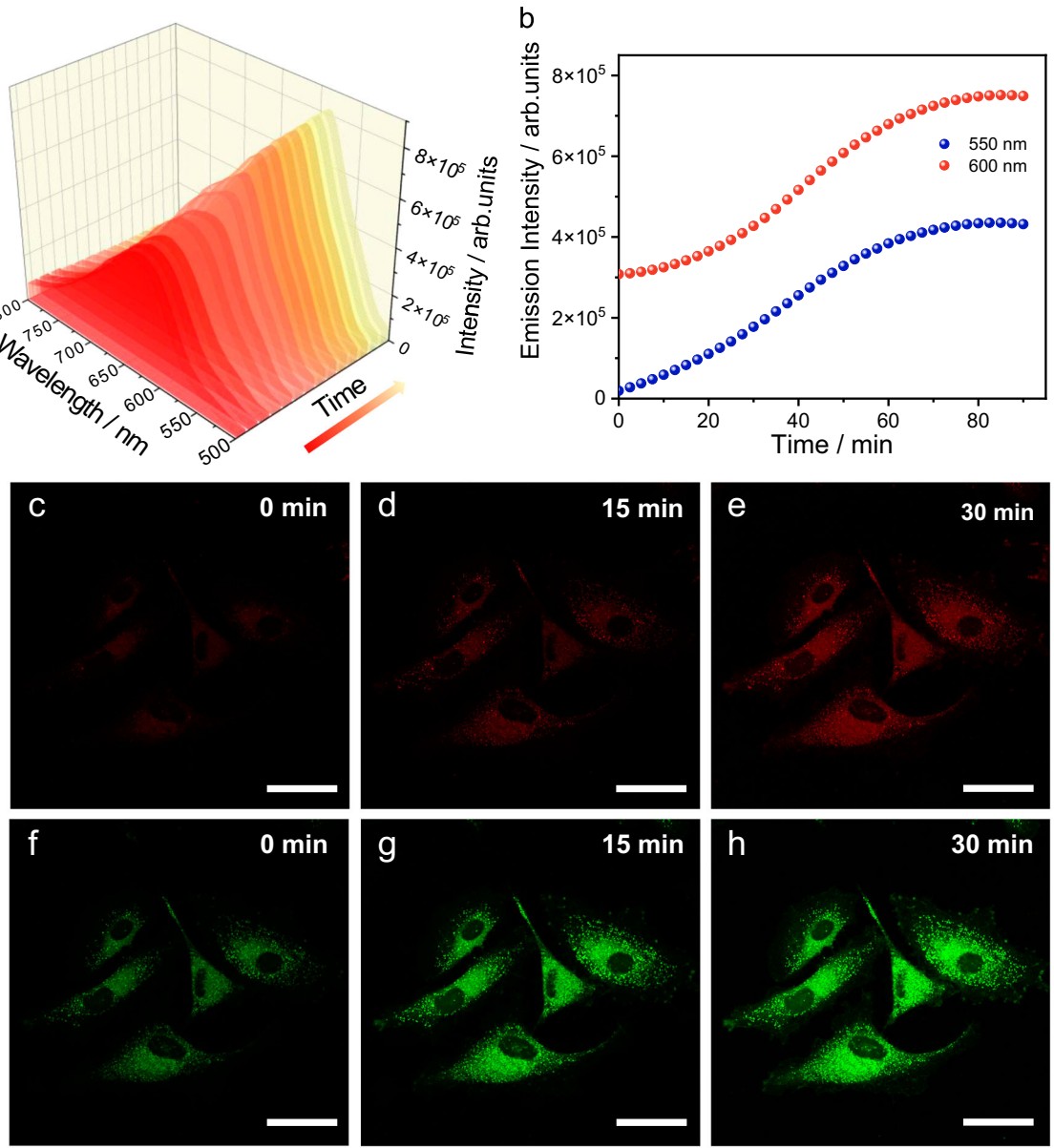

**Fig. 8 | Dynamic cell imaging results. a** Fluorescence spectral variations of **1AggI** ($c = 0.80 \times 10^{-5}$ M) in cell culture medium with 0.05% DMSO. **b** The corresponding kinetic profiles of the conversion from **1AggI** to **1AggII** by monitoring the emission intensities at 550 and 600 nm. $\lambda_{ex} = 405$ nm. CLSM images of the spontaneous transformation of **1AggI** to **1AggII** in A549 cells at (**c**–**e**) red channel ($\lambda_{ex} = 405$ nm, $\lambda_{em} = 600$–700 nm) and (**f**–**h**) green channel ($\lambda_{ex} = 405$ nm, $\lambda_{em} = 500$–599 nm) for 0, 15, and 30 min. Scale bar: 40 μm. Cell imaging was repeated at least three times with similar results.

### Single-crystal growth

Single-crystals of **1**–**4** were performed by the following methods: 5.0 mg of dry **1**–**4** powders were separately put in a small vial where 2 mL of methanol was added. Single-crystal was obtained by slow evaporating the solvent at low temperature for 6 to 7 days.

### DFT and TD-DFT theoretical calculation

The optimized structures of **1AggII₂**, **2AggII₂**, **1AggII₄** and **2AggII₄** were performed on Gaussian 09 software packages[69]. All of the elements were described by the B3LYP/6-31 G(d) basis set. The structures were characterized as a local energy minimum on the potential energy surface by verifying that all the vibrational frequencies were real. The electronic binding energies were defined as products minus reactants. To study the electronic transitions, TD-DFT calculations were performed at the same computational level in water solvent. There are no imagery frequencies for the optimized geometries. The Hirshfeld surfaces and their associated 2D fingerprint plots were carried out by Multiwfn 3.8[70].

### Cell imaging experiments and cytotoxicity tests

All cells were cultured at 37 °C and 5% $CO_2$ in Dulbecco's Modified Eagle's Medium (DMEM), supplemented with 10% fetal bovine serum (FBS) and 1% penicillin-streptomycin solution (P/S) (DMEM-H). The human lung adenocarcinoma epithelial cell line A549 (male, CCL-185) was obtained from the Chinese Academy of Science Cell Bank, and B16-F10 cell line (ATCC, CRL-6475) was purchased from Wuhan Pricella Biotechnology Co., Ltd. The cell lines were authenticated by short tandem repeat profiling. B16-F10 cells were also spot-checked as they express melanin, and cell pellets are black (no other cell lines in use are black when pelleted). All cells were tested negative for mycoplasma contamination. Before the imaging experiments, A549 and B16-F10 cells were incubated with each sample ($c = 0.80 \times 10^{-5}$ M) pre-dissolved in DMEM-H with 0.05% DMSO. Incubation was performed at 37 °C, 5% $CO_2$ for 20 min. The incubated cells were further washed with 400 μL phosphate buffer solution (PBS, pH = 7.4) three times to remove the excess samples.

To evaluate the cell viability, A549 and B16-F10 cells were firstly seeded under 37 °C and 5% CO$_2$ condition for 24 h. No statistical methods were used to predetermine sample size. To clearly see the cell images in the field of vision in the confocal laser scanning microscopy, A549 and B16-F10 cells were seeded at a density of $5 \times 10^3$ cells per well in 96-well plates. After the cell adherence, A549 and B16-F10 cells were incubated with each sample (2 to 10 μmol per 1 mL DMEM-H with 0.1% DMSO) for 24 h. 10 μL MTT solution was added to each well and further incubated for additional 4 h. Absorbance of the samples at 570 nm was measured by an Agilent SpectraMax Paradigm Multi-Mode Microplate Reader. Experiments were performed at least three times in biologically independent replicates. The cell viability was calculated by comparing the absorbance of treated cultures to the absorbance of control cultures.

### Reporting summary
Further information on research design is available in the Nature Portfolio Reporting Summary linked to this article.

## Data availability
The data that support the findings of this study are available in the paper, Supplementary Information, Source data, and also from the authors on request. Crystallographic data have been deposited at the Cambridge Crystallographic Data Centre, under deposition numbers CCDC 2351020 (**1AggII**), 2351018 (**2AggII**), 2368744 (**3AggII**), and 2368743 (**4**). These data can be obtained free of charge from http://www.ccdc.cam.ac.uk/structures/. Source data are provided with this paper.

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

## Acknowledgements

This work was supported by the National Natural Science Foundation of China (22371230, Z.G., 22471219, W.T., and 22071197, W.T.), the Post-doctoral Science Foundation of China (2023M732855, Z.G. and 2022TQ0258, Z.G.), the Shaanxi Fundamental Science Research Project for Chemistry & Biology (22JHQ020, Z.G. and 23JHZ002, W.T.), the Fundamental Research Funds for the Central Universities (G2024KY0605, Z.G. and D5000230114, W.T.), the Open Testing Foundation of the Analytical & Testing Center of Northwestern Polytechnical University (2023T013, Z.G.), and the Innovation Foundation for Master Dissertation of Northwestern Polytechnical University (PF2024026, X.X.).

## Author contributions

Z.G. and W.T. conceived the idea for this project. X.X. and J.Z. performed the experiments, analyzed the data, and produced the artwork under the

direction of Z.G. and W.T. Z.G. and W.T. wrote the paper. H.Y. and W.Y. were involved in data interpretation. All authors contributed to the manuscript preparation.

## Competing interests

The authors declare no competing interests.
