## [Transparent Peer Review file · Nature Communications]

A coopetition-driven strategy of parallel/perpendicular aromatic stacking enabling metastable supramolecular polymerization

Corresponding Author: Professor Wei Tian

Version 0:

Reviewer comments:

Reviewer #1

(Remarks to the Author)

The paper explores a novel non-covalent driving mode for constructing metastable supramolecular polymers through parallel/perpendicular aromatic stacking, which is quite interesting. Time-dependent controllable self-assembly is an attractive research topic, while this manuscript revealed a coopetition-driven strategy through a series of well-designed π -expanded molecular building blocks. Reference compounds and detailed experiments are designed and carried out for demonstrating this constructing strategy to realize the transformation from metastable 0D nanoparticles to thermodynamic 2D nanosheets. Furthermore, it is important that the metastable supramolecular self-assemblies exhibited significant time-dependent polychromatic fluorescence as well as the function of dynamic imaging.

After careful consideration, I suggest this article to be published on Nature Communications after minor revision. There are parts of the article that need to be improved.

1 The solvent affected the aggregation of the polymer and H₂O/DMF was chosen for investigation. Can you explain more about how you determine the optimal composition of H₂O/DMF? Will excessive DMF have a negative effect on its application, such as cell imaging?

2 In Fig 3, the author stated that the EDS mapping showed that the typical elements of C and N were uniformly distributed throughout the sample areas. Can you provide the detailed percentage of element distribution?

3 Demonstration of the designed polymers as π -expanded fluorophores is interesting. I am wondering why A549 cell was chosen? Can those polymers work in a wider range of cellular environments?

4 I am wondering that the intensities of the polymer under various wavelengths did not consistent with the images from Confocal Laser in Fig. 8b. The labeled color is confusing. Could you explain more on this?

5 It is recommended to check the clarity and resolution of all figures and images to ensure they remain high quality. For example, parts of Fig. 2d-f are missing.

Reviewer #2

(Remarks to the Author)

Metastable supramolecular polymerization under kinetic control has recently been recognized as a closer way to biosystems than thermodynamic process. While impressive works on metastable supramolecular systems have been reported, the library of available non-covalent driving modes is still small and a simple yet versatile solution is highly desirable to design for easily regulating the energy landscapes of metastable aggregation. In this nice manuscript, the authors proposed a coopetition-driven metastability strategy for parallel/perpendicular aromatic stacking to construct metastable supramolecular polymers derived from a class of minimalistic monomers consisting of lateral indoles and aromatic core. By subtly increasing the stacking strength of aromatic cores from phenyl to anthryl, the parallel face-to-face stacked aggregates are competitively formed as metastable products, which spontaneously transform into thermodynamically favorable species through the cooperativity of perpendicular edge-to-face stacking and parallel offset stacking. The slow kinetic-to-thermodynamic transformation can be accelerated by adding seeds for realizing the desired living supramolecular polymerization. Besides, this transformation of parallel/perpendicular aromatic stacking accompanied by time-dependent emission change from red to yellow is employed to dynamic cell imaging, largely avoiding the background interferences. The coopetition relationship of

different aromatic stacking for metastable supramolecular systems is expected to serve as an effective strategy towards pathway-controlled functional materials. This manuscript, with good novelty and scientific value, can be published as it is.

Reviewer #3

(Remarks to the Author)

The key findings of this work are (a) pathway complexity with an off-pathway metastable and an on-pathway more stable aggregated states, (b) seeded supramolecular polymerization of the on-pathway assembly from the metastable off-pathway aggregate. Beyond the use of somewhat fancy and needless terminologies borrowed from business school parlance (like coopetition), this work is essentially about supramolecular polymerization under kinetic control and it is hard to find novelty that would warrant its publication in Nat Commun. Even their claim of kinetically controlled supramolecular polymerization in "minimalistic monomers" fails to make an impression.

Finally, the changing fluorescence of the self-assembled system was used to demonstrate dynamic cell imaging. Neither the practical advantage of using a cumbersome dye-aggregate for cellular imaging is clear, nor is there any demonstrable improvement in the image quality.

It is reasonably thorough study that demonstrates what is already known and well-established for close to a decade now. The study neither raises any important questions about pathway complexity, nor reports anything that is particularly surprising or unusual. I cannot recommend its publication in a top-tier journal like Nature Communications.

Version 1:

Reviewer comments:

Reviewer #1

(Remarks to the Author)

I have read through the revised manuscript. I have found that the authors revised this manuscript very carefully according to the reviewers' comments. Thus, I would like to recommend this version of the manuscript for publication.

Response to Referees

Reviewer #1

The paper explores a novel non-covalent driving mode for constructing metastable supramolecular polymers through parallel/perpendicular aromatic stacking, which is quite interesting. Time-dependent controllable self-assembly is an attractive research topic, while this manuscript revealed a coopetition-driven strategy through a series of well-designed π -expanded molecular building blocks. Reference compounds and detailed experiments are designed and carried out for demonstrating this constructing strategy to realize the transformation from metastable 0D nanoparticles to thermodynamic 2D nanosheets. Furthermore, it is important that the metastable supramolecular self-assemblies exhibited significant time-dependent polychromatic fluorescence as well as the function of dynamic imaging.

After careful consideration, I suggest this article to be published on Nature Communications after minor revision. There are parts of the article that need to be improved.

Response: We sincerely appreciate your summary of the manuscript and encouraging comment, and valuable suggestions on our work.

1. The solvent affected the aggregation of the polymer and H₂O/DMF was chosen for investigation. Can you explain more about how you determine the optimal composition of H₂O/DMF? Will excessive DMF have a negative effect on its application, such as cell imaging?

Response: Thanks for your professional comments. We initially screened a variety of solvents to determine their effects on molecular aggregation. Considering the subsequent cell imaging application, H₂O was selected as the poor solvent for the self-assembly. We then retained organic solvents that are miscible with H₂O as potential good solvents and tested the assembly performance of the molecule in these solvents. As shown in Fig. R1 and Supplementary Fig. 11, we found that when IPA, MeOH, THF, Acetone, DMSO, and DMF were used as good solvents during the experiments, the

assembly of **1** all exhibited kinetic pathway complexity. Therefore, considering the boiling points of these solvent mixtures and in order to obtain the aggregated state diagram over a broad temperature range, we finally chose DMF/H₂O as the assembly solvent. The related discussion has been added in the revised manuscript (page 6, marked with a yellow background).

Fig. R1. Time-dependent UV-Vis spectra measurements following the transformation from **1AggI** ($c = 3.00 \times 10^{-5}$ M) to **1AggII** in (a) IPA/H₂O (1 : 24, v/v), (b) MeOH/H₂O (1 : 24, v/v), (c) Acetone/H₂O (1 : 24, v/v), (d) THF/H₂O (1 : 24, v/v), (e) DMSO/H₂O (1 : 24, v/v) and (f) DMF/H₂O (1 : 24, v/v) at 298 K.

For cell imaging, we replaced DMF with DMSO as the good solvent. On one hand, the transformation time from **1AggI** to **1AggII** in DMSO/H₂O is longer than that in DMF/H₂O (Figs. R1e, f), which is more convenient for the incubation in the dynamic cell imaging experiments. On the other hand, DMSO (LD₅₀: 9700–28300 mg/kg, rat through mouth) is less toxic than DMF (LD₅₀: 4000 mg/kg, rat through mouth), DMSO was commonly chosen as the cosolvent for cell imaging (*J. Am. Chem. Soc.* **2021**, *143*, 5396; *J. Am. Chem. Soc.* **2020**, *142*, 18150). Since excessive amounts of DMSO can harm cells, we only added a minimal amount (<0.1%) of DMSO. As shown in Fig. R2 and Supplementary Fig. 52, cytotoxicity tests indicated that at this ratio, the molecule did less damage to cells over a 24-hour period.

Fig. R2. Cytotoxicity tests on A549 cells at different concentrations (0, 2, 4, 6, 8, 10 μM) for **1** after 24 h incubation. The cell viability was assessed via MTT assay. The values presented are the mean \pm SD (n=3).

2. In Fig 3, the author stated that the EDS mapping showed that the typical elements of C and N were uniformly distributed throughout the sample areas. Can you provide the detailed percentage of element distribution?

Response: According to the reviewer's suggestion, the detailed percentage of element distribution from EDS mapping is provided in Fig. R3 and Supplementary Fig. 13. The carbon element accounts for 95.36%, and the nitrogen element accounts for 4.64%. The molecular formula of **1** is $\text{C}_{34}\text{H}_{20}\text{N}_2$, where carbon accounts for 94.44% and nitrogen for 5.56%, closely matching the above experimental data.

2023-10-10 15:06:56 Analysis of spectrum: Spectra from Area #1

Z	Element	Family	Atomic Fraction (%)	Atomic Error (%)	Mass Fraction (%)	Mass Error (%)	Fit error (%)
6	C	K	95.36	6.07	94.63	4.22	3.07
7	N	K	4.64	1.28	5.37	1.46	18.06

Fig. R3. The detailed percentage of element distribution from EDS mapping.

3. Demonstration of the designed polymers as π -expanded fluorophores is interesting. I am wondering why A549 cell was chosen? Can those polymers work in a wider range of cellular environments?

Response: We appreciate your insightful comment on the manuscript. A549 cell is a

common model used for cell imaging to demonstrate the imaging effects in living cells (*Angew. Chem. Int. Ed.* **2024**, *63*, e202409162; *J. Am. Chem. Soc.* **2024**, *146*, 11991; *Chem. Sci.* **2020**, *11*, 8495). Apart from the A549 cells, we have also carried out the cell imaging tests with other common cells, such as B16-F10 cells. The dynamic cell imaging results of **1** in B16-F10 cells were showed in Fig. R4 and Supplementary Fig. 54. The bright emission gradually emerged with increasing time, consistent with the spontaneous transformation kinetics associated with **1AggI** to **1AggII**. In addition, **1** showed negligible toxicity to B16-F10 cells at the cell imaging concentration (Fig. R5 and Supplementary Fig. 55). The related discussion has been added in the revised manuscript (page 17, marked with a yellow background).

Fig. R4. CLSM images of the spontaneous transformation of **1AggI** to **1AggII** in B16-F10 cells at (a)–(c) green channel ($\lambda_{\text{ex}} = 405 \text{ nm}$, $\lambda_{\text{em}} = 500\text{--}599 \text{ nm}$) and (d)–(f) red channel ($\lambda_{\text{ex}} = 405 \text{ nm}$, $\lambda_{\text{em}} = 600\text{--}700 \text{ nm}$) for 0, 15, and 30 min. Scale bar: 40 μm .

Fig. R5. Cytotoxicity tests on B16-F10 cells at different concentrations (0, 2, 4, 6, 8, 10 μM) for **1** after 24 h incubation. The cell viability was assessed via MTT assay. The values presented are the mean \pm SD (n=3).

4. I am wondering that the intensities of the polymer under various wavelengths did not consistent with the images from Confocal Laser in Fig. 8b. The labeled color is confusing. Could you explain more on this?

Response: Thanks for your professional comments. Fig. 8b showed the fluorescence intensity observed in the fluorescence emission spectra. These data were obtained under the same excitation conditions. However, in the confocal laser scanning microscopy images, different light channels use lasers with different power settings, and the gain values for each channel are also different. As a result, the intensity in the images may differs from the intensity observed in the fluorescence spectra. In course of our actual experiments, the gain value set for the green channel ($\lambda_{\text{em}} = 500\text{--}599\text{ nm}$) was higher than that for the red channel ($\lambda_{\text{em}} = 600\text{--}700\text{ nm}$), resulting in a stronger intensity display in the green channel.

5. It is recommended to check the clarity and resolution of all figures and images to ensure they remain high quality. For example, parts of Fig. 2d-f are missing.

Response: We sincerely appreciate your careful review. We have carefully revised the paper, ensuring that all figures have high quality.

Reviewer #2

Metastable supramolecular polymerization under kinetic control has recently been recognized as a closer way to biosystems than thermodynamic process. While impressive works on metastable supramolecular systems have been reported, the library of available non-covalent driving modes is still small and a simple yet versatile solution is highly desirable to design for easily regulating the energy landscapes of metastable aggregation. In this nice manuscript, the authors proposed a coopetition-driven metastability strategy for parallel/perpendicular aromatic stacking to construct metastable supramolecular polymers derived from a class of minimalistic monomers consisting of lateral indoles and aromatic core. By subtly increasing the stacking strength of aromatic cores from phenyl to anthryl, the parallel face-to-face stacked aggregates are competitively formed as metastable products, which spontaneously transform into thermodynamically favorable species through the cooperativity of perpendicular edge-to-face stacking and parallel offset stacking. The slow kinetic-to-thermodynamic transformation can be accelerated by adding seeds for realizing the desired living supramolecular polymerization. Besides, this transformation of parallel/perpendicular aromatic stacking accompanied by time-dependent emission change from red to yellow is employed to dynamic cell imaging, largely avoiding the background interferences. The coopetition relationship of different aromatic stacking for metastable supramolecular systems is expected to serve as an effective strategy towards pathway-controlled functional materials. This manuscript, with good novelty and scientific value, can be published as it is.

Response: We sincerely appreciate your detailed review and positive opinion on our manuscript.

Reviewer #3

The key findings of this work are (a) pathway complexity with an off-pathway metastable and an on-pathway more stable aggregated states, (b) seeded supramolecular polymerization of the on-pathway assembly from the metastable off-pathway aggregate. Beyond the use of somewhat fancy and needless terminologies borrowed from business school parlance (like coopetition), this work is essentially about supramolecular polymerization under kinetic control and it is hard to find novelty that would warrant its publication in Nat Commun. Even their claim of kinetically controlled supramolecular polymerization in "minimalistic monomers" fails to make an impression.

Finally, the changing fluorescence of the self-assembled system was used to demonstrate dynamic cell imaging. Neither the practical advantage of using a cumbersome dye-aggregate for cellular imaging is clear, nor is there any demonstrable improvement in the image quality.

It is reasonably thorough study that demonstrates what is already known and well-established for close to a decade now. The study neither raises any important questions about pathway complexity, nor reports anything that is particularly surprising or unusual. I cannot recommend its publication in a top-tier journal like Nature Communications.

Response: Although the reviewer has different viewpoints on this work, we are still very grateful for the comments.

For the comments of *“The key findings of this work are (a) pathway complexity with an off-pathway metastable and an on-pathway more stable aggregated states, (b) seeded supramolecular polymerization of the on-pathway assembly from the metastable off-pathway aggregate. Beyond the use of somewhat fancy and needless terminologies borrowed from business school parlance (like coopetition), this work is essentially about supramolecular polymerization under kinetic control and it is hard to find novelty that would warrant its publication in Nat Commun. Even their claim of kinetically controlled supramolecular polymerization in "minimalistic monomers" fails to make an*

impression.”, we would like to emphasize the following points. Metastable supramolecular polymerization under kinetic control has recently been recognized as a closer way to biosystem than thermodynamic process (Meijer *et al*, *Nature* **2012**, 481, 492; De Cola *et al*, *Nat. Chem.* **2016**, 8, 10; Würthner *et al*, *Nat. Rev. Chem.* **2020**, 4, 38; Fernández *et al*, *Angew. Chem. Int. Ed.* **2022**, 61, e202203783). So far, the library of available non-covalent driving modes is mainly limited to the teams of intra-/intermolecular hydrogen interactions, as well as a few of hydrogen/halogen interactions, hydrogen/van der Waals interactions, donor/acceptor interactions, and metal/hydrophobic interactions. Besides, sophisticated molecular structures possessing delicate relationship of cooperative, competitive, or orthogonal interactions were regarded as the necessities for achieving pathway complexity. Thus, a simple yet versatile solution is highly desirable to design for easily regulating the energy landscapes of metastable aggregation. Based on these considerations, we focused on the adjustable π -conjugated aromatics, which interact with each other in parallel (offset and face-to-face stacking) and perpendicular (edge-to-face stacking) ways. According to Hunter–Sanders electrostatic model, only offset and edge-to-face stacks are energy minima, whilst face-to-face stacking is rare because of the intrinsic thermodynamic unfavorable nature caused by electrostatic repulsion. Inspired by this, we envisioned that, if a π -conjugated monomer could first be competitively trapped in the manner of face-to-face stacking without any other synergetic interactions, it will spontaneously slide to thermodynamic favored species under the cooperative driving of other two more stable stacking modes. This cooperation coupled with competition driven relationship of parallel/perpendicular aromatic stacking is rare and expected to serve as a simple way for regulating the energy landscapes of metastable aggregation accompanied by variable optical signals. These are our core design ideas, and we have successfully constructed the metastable supramolecular polymers derived from a class of simple π -conjugated monomers by subtly designing the stacking strength of aromatic cores. In order to facilitate understanding and widespread communication, we proposed the “cooperation-driven metastability strategy” to represent this kind of cooperation coupled with competition driven relationship of parallel/perpendicular aromatic

stacking. We believe that this coopetition-driven metastability viewpoints could serve as a promising approach towards synthetic methodology of supramolecular systems and promote the development of new supramolecular functional materials.

For the comments of “*Finally, the changing fluorescence of the self-assembled system was used to demonstrate dynamic cell imaging. Neither the practical advantage of using a cumbersome dye-aggregate for cellular imaging is clear, nor is there any demonstrable improvement in the image quality.*”, we would like to emphasize the practical advantage of the resultant metastable supramolecular polymers for dynamic cell imaging. π -Conjugated fluorophores have been considered as the promising candidates in bioimaging (*Nat. Commun.* **2019**, *10*, 169; *Nat. Commun.* **2021**, *12*, 4993). However, most cell imaging is static, that is, the fluorescent color and intensity are constant, which usually causes the disadvantage of background interferences and possible artefacts. Thus, by virtue of the kinetic-to-thermodynamic transformation accompanying with an emission change from yellow to red, we have applied the designed monomers as the dynamic fluorescent cell imaging agent to rule out artefacts and provide higher background contrast. The CLSM images have showed the obvious fluorescence change at red channel and green channel (Fig. 8 in the main text). Besides, the monomer structure is very simple, and the fluorescence change is spontaneous, which is beneficial to the practical application in dynamic cell imaging field. Lastly, the kinetic-to-thermodynamic transformation has been exploited not only to the dynamic cell imaging, but also to the seed-triggered living supramolecular polymerization. The dynamic cell imaging application is a demonstration tests of structure-function relationship for the metastable supramolecular polymerization.

For the comments of “*It is reasonably thorough study that demonstrates what is already known and well-established for close to a decade now. The study neither raises any important questions about pathway complexity, nor reports anything that is particularly surprising or unusual. I cannot recommend its publication in a top-tier journal like Nature Communications.*”, we would like to point out that, metastable supramolecular polymerization is one of the very important research directions of supramolecular polymerization methodology, which has been developed for more than

ten years, but in order to make metastable supramolecular polymeric system a general synthetic method and to enrich the library of available non-covalent driving forces, a simple yet versatile solution is highly desirable to design for easily regulating the energy landscapes of metastable aggregation. In this manuscript we have reported a simple π -conjugated monomer with novel non-covalent driving modes to achieve this important but difficult-to-obtain supramolecular polymerization. In particular, the highlights of our study are as follows. (1) This is the first demonstration of a co-competition-driven metastability strategy for parallel and perpendicular aromatic stacking, thus enabling the construction of metastable supramolecular polymers with pathway complexity. (2) The kinetic-to-thermodynamic transformation could be exploited not only to the seed-triggered living supramolecular polymerization bypassing the high energy barrier, but also to the dynamic cell imaging excluding the possible artefacts and providing high background contrast. We believe that the current work opens up a new avenue toward both supramolecular chemistry and functional materials.